# Circulating exosome-circRNAs mediated downregulation of FGF9 through ceRNA mechanism aggravates renal fibrosis in diabetic nephropathy

Donglin Yang[1☯], Rongjiang Yin[2☯], Xiaomin Zhang[3☯], Xiaohui Wang[4], Xiaobin Pei[4], Zijie Guo[1], Pengyue Qiao[1], Kehan Zhu[1], Lin Wang[5]*, Pengchao Du[1]*

**1** College of Basic Medical, Binzhou Medical University, Yantai, Shandong, P.R. China, **2** Department of Thoracic Surgery, Yantai Affiliated Hospital of Binzhou Medical University, Yantai, Shandong, P.R. China, **3** Department of Nephrology, Binzhou Medical University Hospital, Binzhou, Shandong, P.R. China, **4** Department of Endocrinology, Yantai Affiliated Hospital of Binzhou Medical University, Yantai, Shandong, P.R. China, **5** Department of Cardiology, Central Hospital Affiliated to Shandong First Medical University, Jinan, Shandong, P.R. China

☯ These authors contributed equally to this work.
* 252983491@qq.com (PD); wanglin54202003@163.com (LW)

## Abstract

Diabetic nephropathy (DN) is one of the most serious microvascular complications of diabetes mellitus. It is characterized by progressive tubulointerstitial fibrosis. The aim of this study was to investigate the role of exosomal circular RNA (circRNAs) in regulating fibroblast growth factor 9 (FGF9) expression in DN through a competitive endogenous RNA (ceRNA) mechanism, and to reveal its potential therapeutic targets. Exosomes were isolated from serum of 3 healthy people and 3 patients with DN by ultra-fast centrifugation method, and the circRNA-miRNA-FGF9 regulatory network was constructed by combining high-throughput circRNA sequencing, bioinformatics analysis and weighted co-expression network (WGCNA). The results showed that the expression of circRNAs in serum exosomes of DN patients was significantly down-regulated, and hsa_circ_0006382 and hsa_circ_0019539 targeted the expression of FGF9 by binding to miR-34a-5p, miR-766-3p, miR-147a and miR-27a-3p. Further verification showed that the expression of FGF9 was decreased in renal tissues of DN patients (AUC = 0.902), and its recombinant protein could inhibit the expression of α-SMA and vimentin in high glucose-induced NRK-52E cells, indicating that activation of the circRNA/miRNA-FGF9 network promotes the EMT of renal tubular epithelial cells. This study revealed for the first time the mechanism of the circRNA-miRNA-FGF9 regulatory network in DN fibrosis, providing a theoretical basis for the development of diagnostic markers and targeted therapy strategies based on exosomal circRNA.

**Data availability statement:** The sequencing data generated in this study have been deposited in the SRA database of the National Center for Biotechnology Information (NCBI), with the accession numbers SRX26678620 (https://www.ncbi.nlm.nih.gov/sra/SRX26678620), SRX26678621 (https://www.ncbi.nlm.nih.gov/sra/SRX26678621), SRX26678622 (https://www.ncbi.nlm.nih.gov/sra/SRX26678622), SRX26678623 (https://www.ncbi.nlm.nih.gov/sra/SRX26678623), SRX26678624 (https://www.ncbi.nlm.nih.gov/sra/SRX26678624), and SRX26678625 (https://www.ncbi.nlm.nih.gov/sra/SRX26678625). Other relevant data are within the manuscript and its Supporting Information files.

**Funding:** The present study was supported by Shandong Natural Science Fund of Shandong Province (grant no. ZR2020MH080); Introduction of Talent Research Initiation Fund of Central Hospital Affiliated to Shandong First Medical University (grant no. YJRC2022017); The Projects of Medical and Health Technology Development Program in Shandong Province (grant no. 202003050666, 2019WS310); The Projects of Technological Innovation Development Program in Yantai City (grant no.2021YD060, 2022YD071); Traditional Chinese medicine science and technology project of Shandong Province (grant no. M-2023017); Clinical +X project of Binzhou Medical University (BY2021LCX24). The funders had no role in study design, data collection and analysis, decision to publish, or preparation of the manuscript.

**Competing interests:** The authors have declared that no competing interests exist.

## Introduction

Diabetic nephropathy (DN) is one of the microvascular complications of diabetes and the main cause of end-stage renal disease [1]. Progressive tubulointerstitial fibrosis is the key pathological change observed in DN. The pathophysiological mechanism of DN is complex, including immune regulation, inflammation, oxidative stress and other mechanisms. However, effective treatments which are able to reduce or even reverse tubulointerstitial fibrosis are limited. Therefore, it is of utmost urgency to identify the therapeutic targets of DN to delay or reverse its progression.

Exosomes originate in the extracellular vesicles of endosome and are the most typical type of extracellular EVs [2,3]. Exosomes can be isolated from a variety of mammalian cells and are present in almost all types of biological fluids, including blood, urine, prostate fluid and saliva [4,5]. Exosomes contain a variety of signaling molecules, including proteins, DNA, mRNAs, miRNAs, long-chain non-coding RNAs, circRNAs, etc. They are the key mediators of paracrine function, and can fuse with the cell membrane, transfer nucleic acid substances and proteins to target cells, and mediate functions in cells [6,7]. circRNAs are a class of non-coding RNAs (ncRNAs) produced by reverse splicing, which are highly stable in structure and evolutionarily conserved, and are stably enriched in exosomes. Exosomal circular RNAs (circRNAs) have the dual characteristics of exosome delivery function and circRNA transcriptional regulation, and have the potential to become DN biomarkers due to their high stability, expression specificity and easy detection. circRNAs can function as sponges for miRNAs, indirectly regulating mRNA expression, known as competitive endogenous RNA (ceRNA) function [8]. The dysregulation of ceRNAs disrupts the balance of cellular processes and functions. circRNAs have become key regulators of a number of diseases, including cancer, cardiovascular disease, kidney disease etc. [9,10]. Studies have shown that circRNAs have been detected in the blood of patients with DN [11]. However, studies on the role of circulating exosomes-circRNAs in DN are limited.

Fibroblast growth factor (FGF) is one of the most diverse groups of growth factors in vertebrates. FGFs plays a key role in metabolism by regulating the kidneys, liver, brain, gut and fatty tissue. FGF9, also known as glial activator, belongs to the FGF9 subfamily. FGF9 has a high affinity for FGFR3 and FGFR2 [12], and is involved in the growth and development of the lungs, kidneys and bone tissue [13,14]. Among the growth factors known to be expressed in embryonic kidneys, FGF9 has been found to promote the proliferation of nephron progenitor cells *in vitro* [15]. It has been shown that the knockout of FGF9 and FGF20 alone or in combination leads to the apoptosis of nephron progenitors and subsequent renal stunting [16]. However, the effects of FGF9 on DN remain unclear.

Exosome circRNA is associated with inflammatory mechanism [17]. Besides, type 2 DM [18] and diabetic complications such as diabetic kidney disease [19–25] and diabetic neuropathy [26] are also characterized with chronic inflammatory burden. Hence, studying exosome-circRNAs mediated downregulation of FGF9 through ceRNA mechanism in DN is reasonable. In the present study,

a circRNA-miRNA-FGF9 regulatory network was constructed by the high-throughput sequencing of serum circRNAs, miRNA binding site prediction, and the WGCNA co-expression of DN, and the role of FGF9 in the epithelial-mesenchymal transition (EMT) of renal tubular epithelial cells was further verified by *in vitro* cell experiments. A graphical abstract of the present study was illustrated in (Fig 1).

## Materials and methods

### Human sample inclusion principle

All human specimens were obtained from the Affiliated Hospital of Binzhou Medical University from April 22, 2021 to March 11, 2023. The patients were provided written informed consent to participate in this study. The six serum samples were from three healthy people and three DN patients. The criteria for inclusion of healthy people were disease-free. All experiments were all pathologically proven with an estimated glomerular filtration rate (eGFR) no less than 30 ml/min/ 1.73 m$^2$ (based on the Chronic Kidney Disease Epidemiology Collaboration (CKD-EPI) equation) [27]. All experiments involving human subjects were approved by the Ethics Committee of Binzhou Medical University (protocol no. 2020-405).

### Exosome isolation

The purified exosomes were obtained from serum by ultrafast centrifugation. Specific process: The sample was transferred to a new centrifuge tube at 37°C, centrifuged at 2000 × g at 4°C for 30 min. Then the supernatant was carefully transferred to a new centrifuge tube at 10000 × g, 4°C, 45 min, and centrifuged again to remove the large vesicles. Filter the supernatant with 0.45 μm filter membrane and collect the filtrate. The filtrate was transferred to a new centrifuge tube, the overspeed rotor was selected, and centrifuged at 100,000 × g at 4°C for 70 min. After removing the supernatant and re-suspension with 10 mL pre-cooling 1 × PBS, the overspeed rotor was selected and centrifuged again for 70 min at 4°C, 100,000 × g. Finally, the supernatant was pre-cooled with 100 μL 1 × PBS, 20 μL electron microscopy, 10 μL particle size, re-suspension, and the remaining exosomes were stored at −80°C for subsequent sequencing [28].

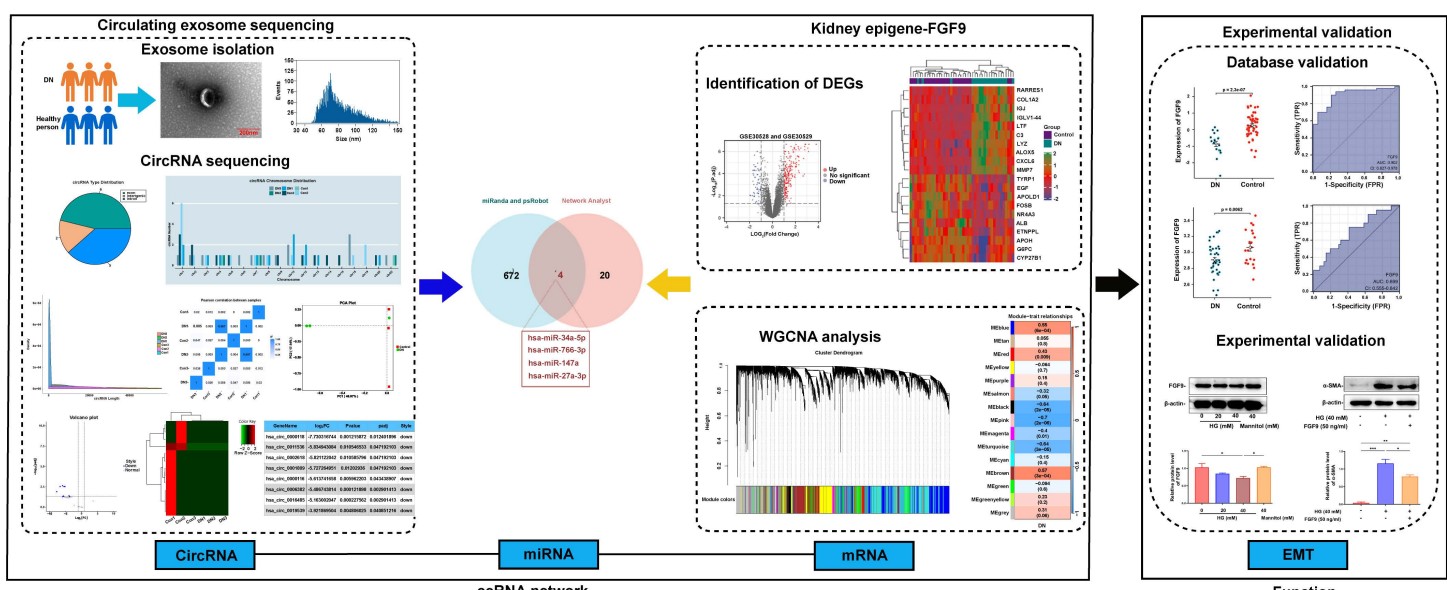

**Fig 1. Graphic abstract of the study.**

### Transmission electron microscopy (TEM)

10µL exosomes were absorbed, deposited on the copper net for 1 min, and the floating liquid was absorbed by filter paper. Then 10 µL uranyl acetate was added to the copper mesh and precipitated for 1 min. The filter paper absorbed the float. After drying at room temperature for a few minutes, electron microscope imaging (HT-7700 Hitachi microscope, Hitachi, Tokyo, Japan).

### Nanoparticle tracking analysis (NTA)

Test according to manufacturer's instructions. After the standard sample had been tested for instrument performance and found to be qualified. The 30 µL exosomes were extracted for particle size analysis.

### Exosome- circRNAs sequencing

After RNA extraction, RNA detection, rRNA and linear RNA removal, sequencing library construction, and passing the library inspection, Illumina PE150 sequencing was performed after pooling different libraries according to the effective concentration and target data volume requirements. The transcriptome sequencing Reads were compared with reference genome using Hierarchical Indexing for Spliced Alignment of Transcripts (HISAT2) software, and circRNA was screened [29]. The login number of the data set: PRJNA1184123.

### Differential expression of circRNAs

Two circRNA prediction software find_circ [30] and CIRI2 [31] were used to predict and screen circRNAs. Reads Count values of different circRNAs were counted according to the comparison results of each sample. The calculated transcripts per million (TPM) values were used to normalize expression levels, and dimensionality of the data was reduced by principal component analysis (PCA) [32]. According to the set threshold: $| \log_2(\text{Fold Change, FC}) | > 1$ and adjusted $P$- ($Padj$-) value $< 0.05$, circRNA differences between samples were screened. The corresponding Fold Change and $Padj$- value of up-regulated and down-regulated genes were obtained, and the volcano map was used for visualization. According to the TPM values of the different circRNAs, the clustering heat map between the samples was drawn.

### CircRNA-miRNA regulatory network analysis

The miRNA-specific binding sites of differentially expressed circRNAs were predicted using miRanda [33] and psRobot [34]. Based on the binding site relationship between miRNAs and circRNAs, Cytoscape software was used to screen the differentially expressed circRNAs and binding information [35]. The visualization of circRNA-miRNA regulatory network was carried out.

### Patient datasets

Six GEO datasets related to DN were screened: GSE30528 (GPL571) included 9 DN cases and 13 controls; GSE30529 (GPL571) included 10 DN cases and 22 control cases. The matrix data of GSE30528 and GSE30529 were combined and normalized using the "sva" package in R software for differential gene analysis. GSE30122 (GPL571) included 19 DN cases and 50 control cases. GSE96804 (GPL17586) included 41 DN cases and 20 control cases. As an external validation data set. GSE142025 (GPL20301) included 27 DN cases and 9 controls for weighted correlation network analysis (WGCNA).

### Identification of Differential Expression Analysis

PCA was performed to observe the distribution among the groups. The "limma" software package in R was used to identify differential expression genes (DEGs) between normal and DN samples, $Padj < 0.05$ and $|\log_2 FC| > 1$ gene was DEGs.

 

The "Pheatmap" R software package and "ggplot2" R software package were used to draw the heatmap and volcano map of DEGs respectively.

## WGCNA co-expression Analysis

The gene cluster co-expression network was constructed by the "WGCNA" R software package. Select an appropriate soft threshold using the pickSoftThreshold function in the WGCNA software package. Each gene was divided into modules according to the weighted correlation coefficient, and then the correlation analysis between modules and traits and between modules was carried out, and finally epigenes were screened.

## Functional enrichment analysis

GO and KEGG pathway analysis was performed using the "clusterProfiler" R software package to evaluate gene-related biological processes (BP), molecular functions (MF), cell components (CC), and gene-related signaling pathways.

## Protein -protein interaction (PPI) network analysis construction

The STRING database (http://string-db.org/) [36] and Cytoscape [37] were used for protein-protein interaction (PPI) analysis and modular analysis of differential genes. Import DEGs into STRING database and analyze PPI.

## ROC evaluation

Use the "pROC" R software package to construct the ROC curve and calculate the area under the curve (AUC). The higher the AUC, the higher the diagnostic value of this gene.

## Cell culture and treatment

Normal rat kidney-52 epithelial (NRK-52E) were cultured in 1640 medium containing 10% fetal bovine serum and 1% penicillin-streptomycin mixture in a 5% $CO_2$ incubator at 37°C. NRK-52E cells were stimulated by the high-concentration glucose (HG, 20–40 mM) and the same concentration of mannitol as osmolarity control. After HG stimulation, FGF9 recombinant protein (50 ng/ml) treatment followed.

## CCK-8 tests

In the CCK-8 experiment, NRK-52E cells were planted in 96-well plates ($5 \times 10^3$ cells/well), incubated for 48 h. Replaced the medium with CCK8 working solution, and continued to incubate in 37°C for 4 h. Finally, the absorbance at 450 nm was measured with an enzyme-labeled instrument (Instrument models: SpectraMax Mini).

## Western blot assays

Cell homogenates were lysed in an ice-cold radioimmunoprecipitation (RIPA, Beyotime, Beijing, China) lysis buffer with 1 mM of phenylmethanesulphonyl fluoride (PMSF, Beyotime, Beijing, China), and then were seperated by sodium dodecyl sulfate polyacrylamide gel electrophoresis (SDS-PAGE) and transferred to polyvinylidene fluoride membranes (PVDF, Millipore, Bedford, MA). And then the membranes were incubated with primary antibodies at 4°C overnight, washed, and incubated with secondary antibodies. Finally, the blots were visualized by an enhanced chemiluminescence detection system (Tanon, Shanghai, China). Primary antibodies included those against FGF9 (1:1000, product number: A6374, ABclonal), α-SMA (1:1000, product number: #19245T, Cell Signaling Technology, Inc.), vimentin (1:4000, product number: 10366-1-AP, ProteinTech Group, Inc.) and β-actin (product number: 60008-1-Ig, ProteinTech Group, Inc.). Second antibodies included HRP-conjugated Affinipure Goat Anti-Mouse IgG (H + L), (1:5000, product number: SA00001-1, ProteinTech Group, Inc.), HRP-conjugated Affinipure Goat Anti-Rabbit IgG (H + L), (1:10000, product number: SA00001-2, ProteinTech Group, Inc.).

## Immunofluorescence of α-SMA

NRK-52E cells were seeded with a density of $10^6$ cells/ well and incubated at 37°C, 5% $CO_2$ for 24h. The cells were fixed with 4% paraformaldehyde, permeabilized with 0.1% Triton X-100, and blocked with nonspecific antigens with 3% BSA. The cells were incubated with α-SMA antibody (1:200, product number: #19245T, Cell Signaling Technology, Inc.) overnight at 4°C, followed by Anti-rabbit lgG Fab 2 Alexa Fluor 488 (Cat. #4412, 1:200, Cell Signaling Technology, Inc.) at room tem-perature for a further 2h, and the fluorescent nuclear DAPI stain (Beyotime, Beijing, China) for 5min. Then fluorescent images were observed under a confocal laser scanning microscope (ZEISS, Germany).

## Statistical analysis

Data were expressed as means ± s.e.m. The significance of the differences in mean values between multiple groups were examined by one-way analysis of variance via GraphPad Prism 9.4 software. $P < 0.05$ was considered statistically significant.

## Results

### Expression of serum circRNAs in patients with DN

Serum was obtained from 3 healthy individuals and 3 patients with DN, and exosomes were isolated and purified using the ultrafast centrifugation method. Transmission electron microscopy (TEM) was used to observe spherical particles with a diameter of 30–150 nm (Fig 2A). The size of the particles analyzed by nanoparticle tracking analysis (NTA) technology was 81.35nm and the concentration was $1.93 \times 10^9$ particles/ml (Fig 2B). The exosomes were extracted successfully. The purified exosomes were sequenced by circRNAs. The length distribution of circRNAs in each sample was statistically analyzed and plotted, the results showed that the length of most circRNAs was distributed between 149–15149nt (Fig 2C). The type distribution map of the source regions of circRNAs showed circRNAs were derived from the splicing of exons, introns and intergenic regions (Fig 2D). The chromosomal distribution map showed that the circRNA host genes were located on chromosome 1, 10, and 18. (Fig 2E). The results of sample correlation heat map showed that the correlation between DN1 and DN2 was 0.997, indicating that the two samples were highly similar (Fig 2F). The results of PCA diagram showed that the control group (red) and DN (blue) clustered on both sides of PC1 (46.97%), indicating that circRNA expression was significantly different between the two groups (Fig 2G). The differential expression of circRNAs was analyzed, and the volcano map revealed that the expression of differentially expressed circRNAs was significantly decreased in DN (Fig 2H). The clustering heatmap analysis revealed that circRNAs were generally green (low expression) in DN and red (high expression) in control group, indicating that low expression of circRNA might be related to DN (Fig 2I). The top eight circRNAs were listed in descending order according to the | $log_2FC$ | value (Fig 2J).

### Functional enrichment analysis of source genes with differentially expressed circRNAs

In order to further examine the biological function of differentially expressed circRNAs, GO, KEGG and Reactome were used to enrich the source genes of differentially expressed circleRNAs. Cellular component (CC) enrichment analysis revealed that these source genes were related to the N-terminal protein acetyltransferase complex, the site of DNA damage, the site of double-strand break, etc. Molecular function (MF) enrichment analysis revealed that these source genes were related to ATPase activator activity, RNA stem-loop binding, alpha-mannosidase activity, etc. (Fig 3A). Similarly, KEGG pathway analysis revealed enrichment in protein processing in the endoplasmic reticulum (*Padj* < 0.01), porphyrin and chlorophyll metabolism (*Padj* < 0.05), N-Glycan biosynthesis (*Padj* < 0.05), Aminoacyl-tRNA biosynthesis (*Padj* < 0.05). (Fig 3B). The results of Reactome enrichment analysis revealed that the circRNAs were related to tRNA modification in the nucleus and cytoplasm, and tRNA aminoacylation (*Padj* < 0.05), etc. (Fig 3C).

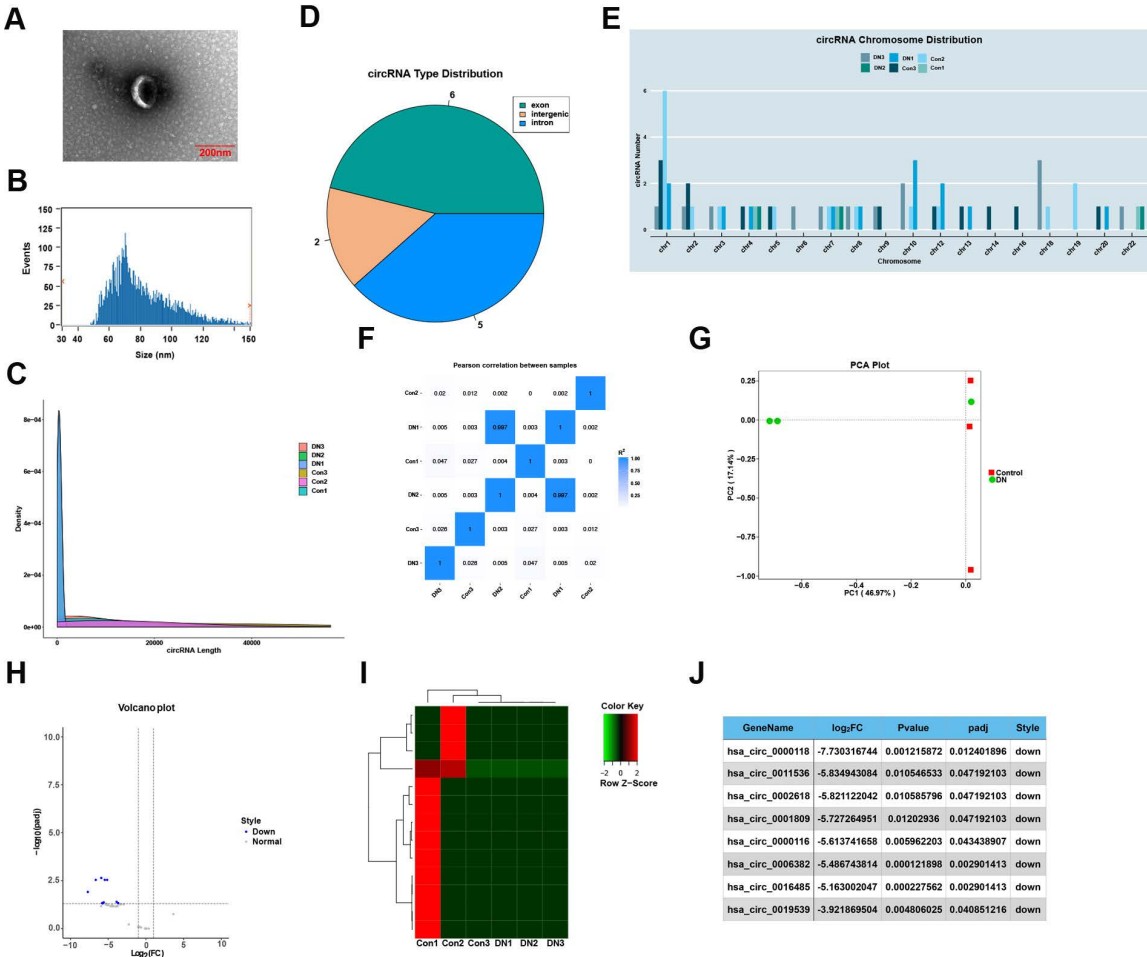

**Fig 2. Expression of serum circRNAs in patients with DN.** (A) Representative images of exosome morphology by transmission electron microscopy. (B) The vesicle size distribution of a representative exosome was measured by nanoparticle tracking analysis (NTA). (C) CircRNA length distribution statistics. (D) CircRNA source distribution diagram. (E) CircRNA chromosome distribution. (F) Sample correlation heat map. (G) Principal component analysis (PCA) diagram. (H) Volcanic plot of different circRNAs. (I) Heat map of differential circRNAs expression profiles. (J) The first 8 meaningful differences presented by circRNAs.

## Identification of DEGs

The GSE30528 and GSE30529 dataset samples were combined after batch effect removal, including 19 DN tubule samples and 25 control samples. The data box plot after processing revealed that the median of each sample was almost on the same line, which meant the data were standardized successfully (Fig 4A). PCA results showed PC1 (21%) and PC2 (16.9%), which indicated that samples were clustered according to DN and control group (Fig 4B). The volcano map showed 202 upregulated genes and 64 downregulated genes in DN according to the analysis of DEGs (Fig 4C). The heatmaps showed the top 10 upregulated (RARRES1, COL1A2, IGJ, IGLV1–44, LTF, C3, LYZ, ALOX5, CXCL6, MMP7) and downregulated genes (TYRP1, EGF, APOLD1, FOSB, NR4A3, ALB, ETNPPL, APOH, G6PC, CYP27B1) in DN (Fig 4D).

## WGCNA identifies eigengenes associated with DN

In order to ensure the scale-free topology and high fit of the co-representation network, the optimal soft threshold was determined to be 12 (Fig 5A). The weighted co-expression network was constructed, and the hierarchical clustering tree of genes

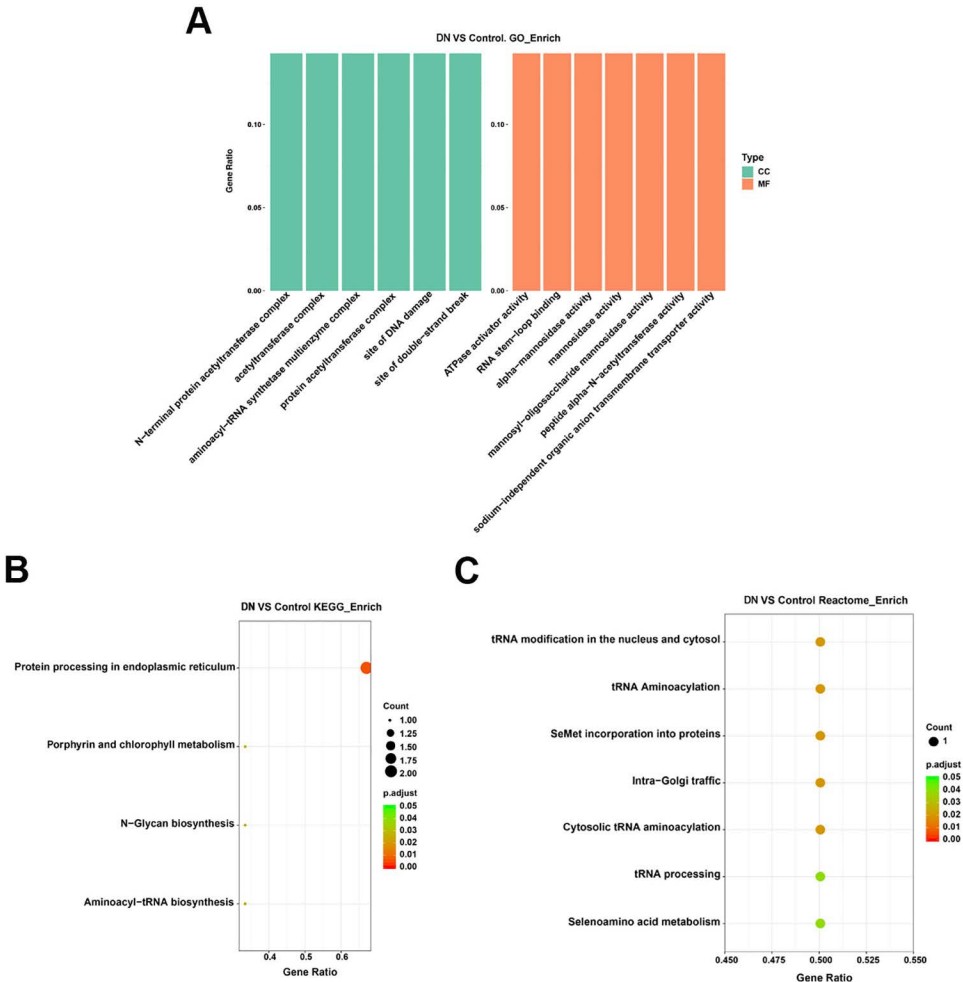

**Fig 3. Biological functional enrichment of differentially derived circRNA genes.** (A) GO Bubble Chart enrichment analysis of differentially circRNAs derived genes. (B) KEGG enrichment analysis of differentially derived circRNAs genes. (C) Reactome enrichment analysis of differentially derived circRNAs genes.

was drawn. Genes clustered to the same branch were divided into the same module, and different colors represented different modules (Fig 5B). The correlation and significance between modules and DN were calculated and the heatmap of the correlation between modules and traits was drawn. The results revealed that 15 gene modules with similar co-expression traits were found, among which the pink module, turquoise module and black module had the highest correlation with DN traits and exhibited a negative correlation (Fig 5C). Further analysis of the correlation between gene membership (MM) and gene significance (GS) in the three modules revaled the positive correlation between MM and GS, in the pink module (MM and GS correlation=0.34; $P$=3E-07, Fig 5D), in the turquoise module (MM and GS correlation=0.51; $P$<1e-200, Fig 5E) and the black module (MM and GS correlation=0.67; $P$=1.1e-49, Fig 5F), indicating that the more important genes in the module played a more significant role in DN. A total of 845 eigengenes were screened according to the criteria of |MM|>0.8 and |GS|>0.6.

### FGF9 is the key gene of DN

A total of 34 common genes were obtained by the intersection of eigengenes and DEGs (Fig 6A). A PPI network of 34 common genes was constructed, and 34 nodes and 67 edges were identified (PPI enrichment $P$-value<1.01e-10) (Fig

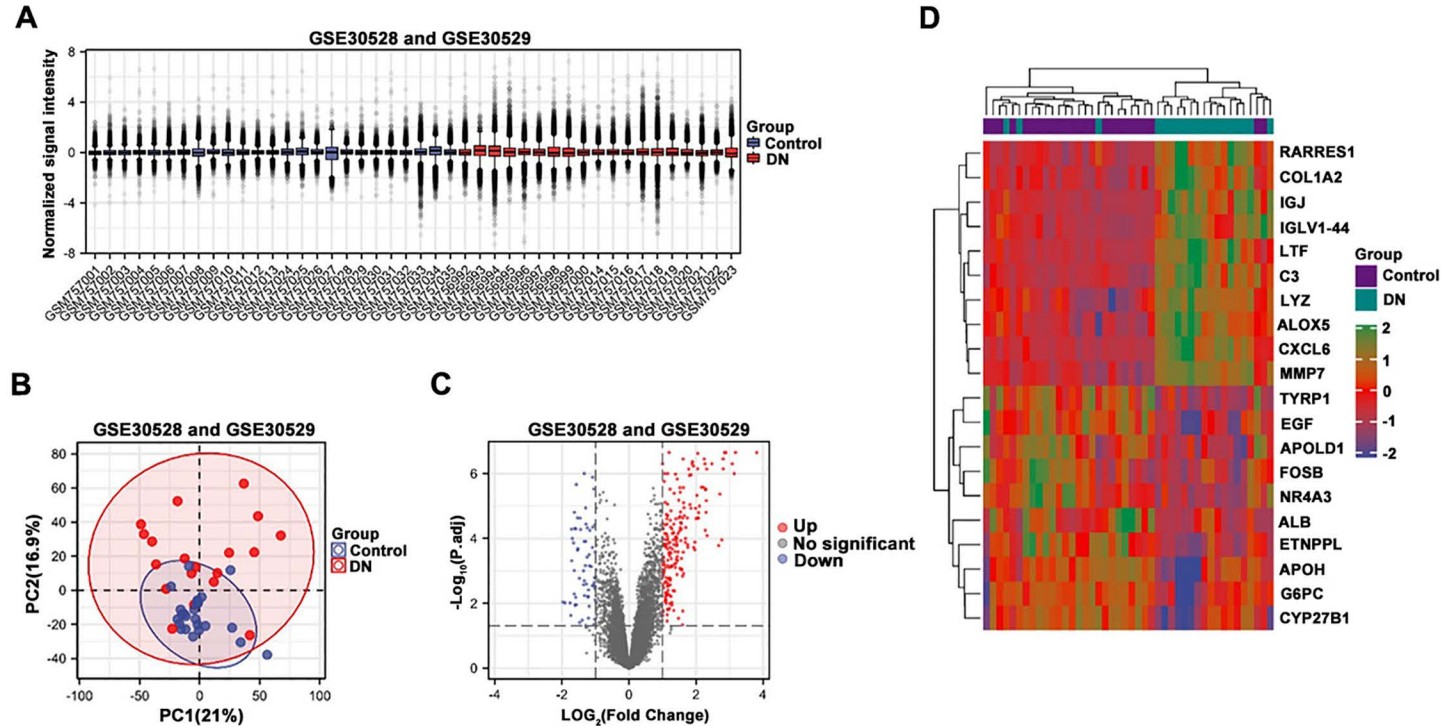

**Fig 4. Identification of differential expression analysis.** (A) Boxplot of the combined data of the GSE30528 database and the GSE30529 database sample after batch-effect removal. (B) PCA diagram of GSE30528 database and GSE30529 database after data merger. (C) Volcanic plot of differential genes. (D) Heat map of the expression profiles of the first 10 up-regulated and down-regulated genes were taken.

6B). The thicker the cable, the higher the reliability of the cable. The larger the node, the greater the significance of the protein in the key biological processes of DN. Among the negatively correlated genes, the FGF9 node was the largest; thus, FGF9 was considered to be the key gene of DN. The functional enrichment analysis of 34 common genes was performed. The results of KEGG analysis revealed that they were significantly associated with the PI3K-AKT signaling pathway, Rap1 signaling pathway and AGE-RAGE signaling pathway in diabetic complications (Fig 6C). The results of GO analysis revealed that they were closely related to extracellular matrix structural constituent, collagen-containing extra-cellular matrix and smad binding (Fig 6D). The Network Analyst 3.0 online network tool was used to predict the miRNAs for FGF9. The mRNA-miRNA co-expression network based on the association between mRNAs and miRNAs was con-structed using Cytoscape (Fig 6E).

### Construction of the circRNA/miRNA-FGF9 regulatory network

The ceRNA mechanism of circRNA/miRNA is a key strategy for determining the function of circRNAs. Bioinformatic analysis using the miRanda and psRobot databases identified 676 miRNAs that may interact with differentially expressed circRNAs. The top 10 miRNAs are presented in order from the highest to lowest Max-Score (Fig 7A). The visualization of the circRNA/miRNA interaction network was performed using Cytoscape (Fig 7B). The intersection of 676 miRNAs with 24 FGF9 target miRNAs yielded four common miRNAs, namely, hsa-miR-34a-5p, hsa-miR-766-3p, hsa-miR-147a and hsa-miR-27a-3p (Fig 7C). Their circRNAs were hsa-circ-0006382 and hsa-circ-0019539, respectively, and the circRNA/miRNA interaction network was visualized using Cytoscape (Fig 7D).

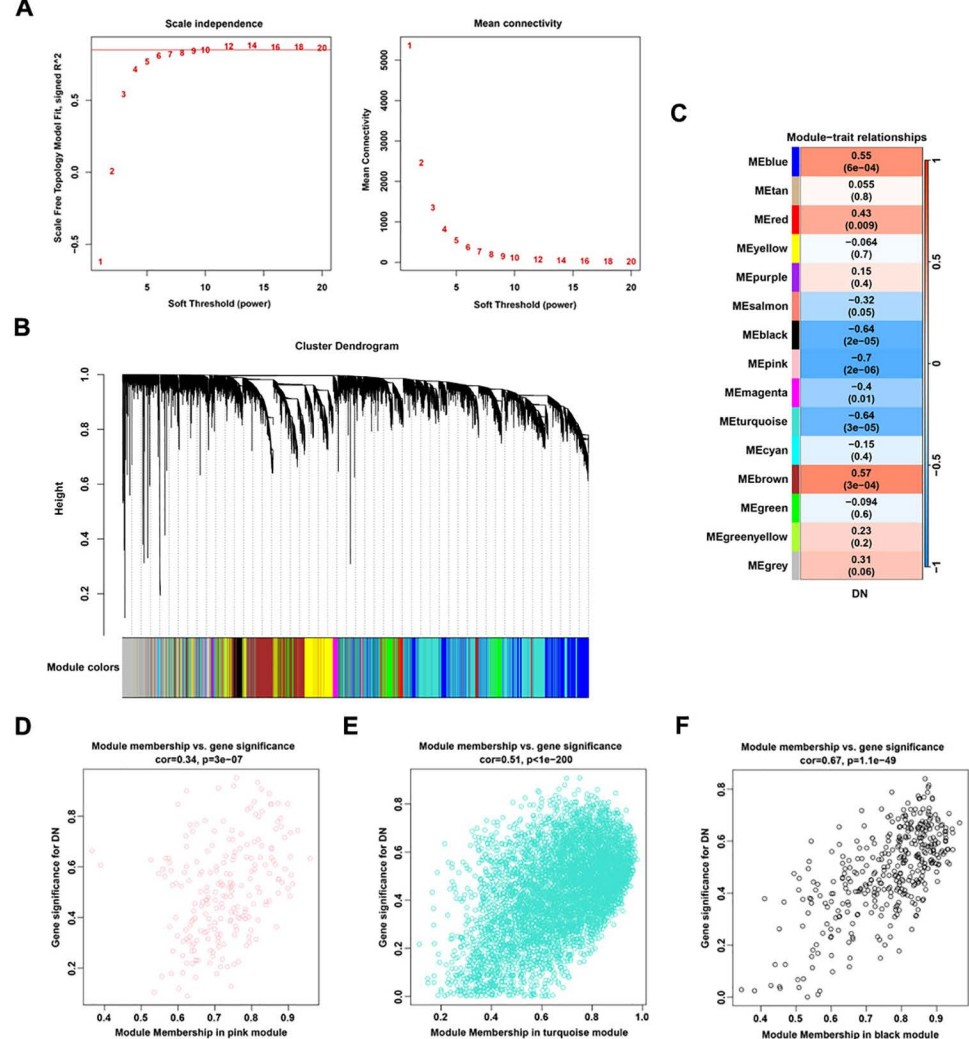

**Fig 5. Weighted Correlation Network Analysis to identify DN related feature genes.** (A) The optimal soft threshold was determined to be 12. (B) Hierarchical clustering tree shows different modules related to DN. (C) Heat map of correlation between characteristic genes of different modules and DN. (D) Scatter plot of correlation between Gene and Module Membership (MM) and Gene Significance (GS) in pink module. (E) Scatter plot of correlation between turquoise module MM and GS. (F) Scatter plot of correlation between black module membership and DN gene traits.

### FGF9 external database validation and functional analysis

The expression of FGF9 was verified using the DN kidney tissue GSE30122 and GSE96804 datasets. The results revealed that the expression of FGF9 in the kidney tissues of patients with DN in both the GSE30122 and GSE96804 datasets was significantly decreased (Fig 8A and B). ROC curves were further drawn to verify the diagnostic value of FGF9. The AUC values of FGF9 in the GSE30122 and GSE96804 datasets were 0.902 and 0.699, respectively (Fig 8C and D), indicating that FGF9 had a high diagnostic value for DN. The function of FGF9 in renal tubular epithelial cells was further verified using cell experiments *in vitro*. The results revealed that the expression of FGF9 in NRK-52E cells was significantly decreased under high glucose stimulation in a concentration-dependent manner (Fig 8E and F). In NRK-52E cells, the expression of α-SMA and vimentin was significantly decreased after FGF9 treatment. (Fig 8G–J). The CCK8 test showed that, compared with control, NRK cell viability decreased significantly after 48h of high glucose stimulation, which

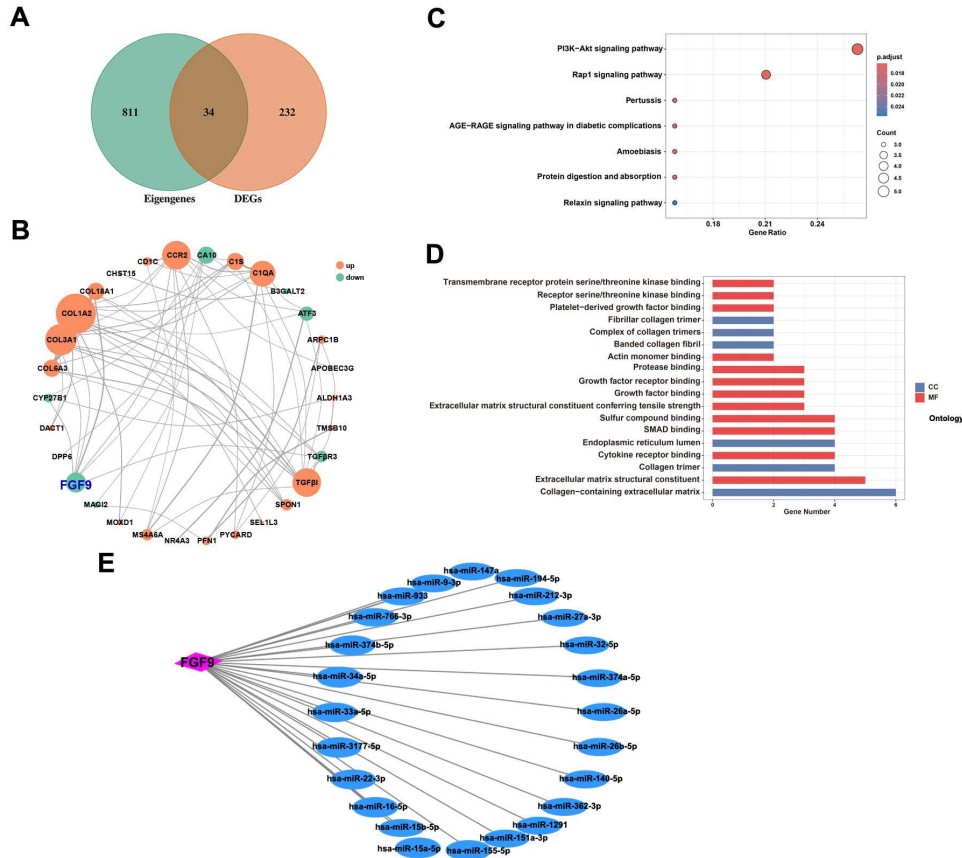

**Fig 6. Central gene screening and functional enrichment analysis.** (A) Venn diagram of intersection of DEGs and characteristic genes. (B) Protein Interaction (PPI) networks of 34 differentially expressed proteins. (C) GO Bubble Chart of differentially expressed genes. (D) KEGG enrichment of differentially expressed genes. (E) Network regulatory diagram of FGF9 target miRNAs predicted by Network Analyst.

was significantly improved by FGF9 intervention for 48h (Fig 8K). Consistently, immunofluorescence staining revealed that a-SMA expression level was obviously increased in HG-treated cells compared with normal glucose, a-SMA expression decreased in NRK-52 cells of HG+FGF9 group (Fig 8L), indicating that FGF9 could alleviate EMT in NRK-52E cells.

## Discussion

In the present study, circRNA profiles in serum-derived exosomes of DN were revealed by circRNA high-throughput sequencing, which clarified the differential expression of circulating exosome-circRNAs in patients with DN and healthy individuals, and predicted the binding sites of miRNAs to search for downstream mRNA. At the same time, bioinformatics analysis and WGCNA co-expression analysis were used to screen the renal tissue-specific gene, FGF9, in DN, predict the miRNA binding site of FGF9, and the intersection with differential circRNAs to construct the circRNA-miRNA-FGF9 regulatory network. Circulating exosome-circ-0006382/-circ-0019539-FGF9 was found to promote hyperglycemia-induced renal tubular cell fibrosis.

circRNAs are a novel type of RNA which is different from traditional linear RNA. circRNAs have a closed ring structure and exist in the eukaryotic transcriptome in large numbers [38,39]. The dysregulation of circRNA expression under pathological conditions promotes the pathogenesis and progression of various diseases, including kidney disease. In recent years, there has been increasing interest in the role of circRNAs in kidney diseases, including renal cell carcinoma, acute

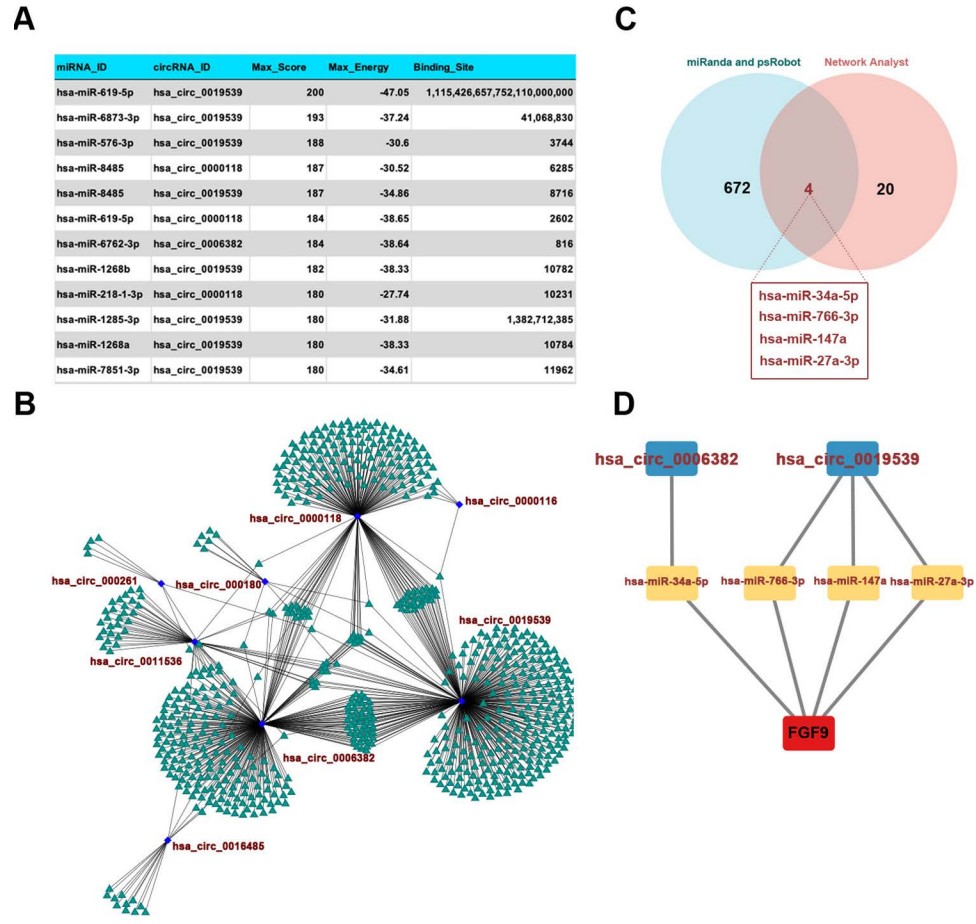

**Fig 7. Construction of circRNA-miRNA-mRNA regulatory network.** (A) The top ten miRNAs were displayed in order of Max-Score from higher to lower for differentially expressed circRNAs and miRNAs binding sites. (B) Regulatory network diagram of differentially expressed circRNAs and their predicted miRNAs. (C) Venn diagram of common miRNAs predicted by Network Analyst and miRanda and psRobot. (D) CeRNA regulatory network diagram of the four common miRNAs and their corresponding serum exosomes circRNAs and FGF9.

kidney injury, DN and other glomerular diseases [11,40,41]. Studies have shown that exosome hsa_circ_0125310 promotes cell proliferation and fibrosis in DN by sponging miR-422a and targeting the IGF1R/p38 axis [42]. Exosome circ_DLGAP4 promotes the progression of DN by absorbing miR-143 and targeting the ERBB3/NF-KB/MMP-2 axis [43]. In the present study, by comparing the circRNA expression profiles in healthy individuals and patients with DN, it was found that the expression of differentially expressed circRNAs in DN was significantly reduced. The majority of these dysregulated circRNAs originate from exons or introns. It has been reported that both exon circRNAs and intron circRNAs may have potential functions in gene regulation [8]. The biological functions of differentially expressed circRNAs were further studied, and it was found that they were related to endoplasmic reticulum stress and DNA damage. Endoplasmic reticulum stress is closely related to the development of DN [44].

Exosome-circRNAs function as sponges for miRNAs, competitively binding to miRNAs, thereby reducing the inhibitory effect of miRNAs on their target mRNAs, which is termed the ceRNA mechanism [45]. Research has revealed the role of circRNAs in DN. The deletion of circRNA_010383 promotes the expression of TRPC1 protein through the sponge of miRNA-135a, resulting in the formation of DN proteinuria, mesangial cell proliferation and renal fibrosis [46]. Another study

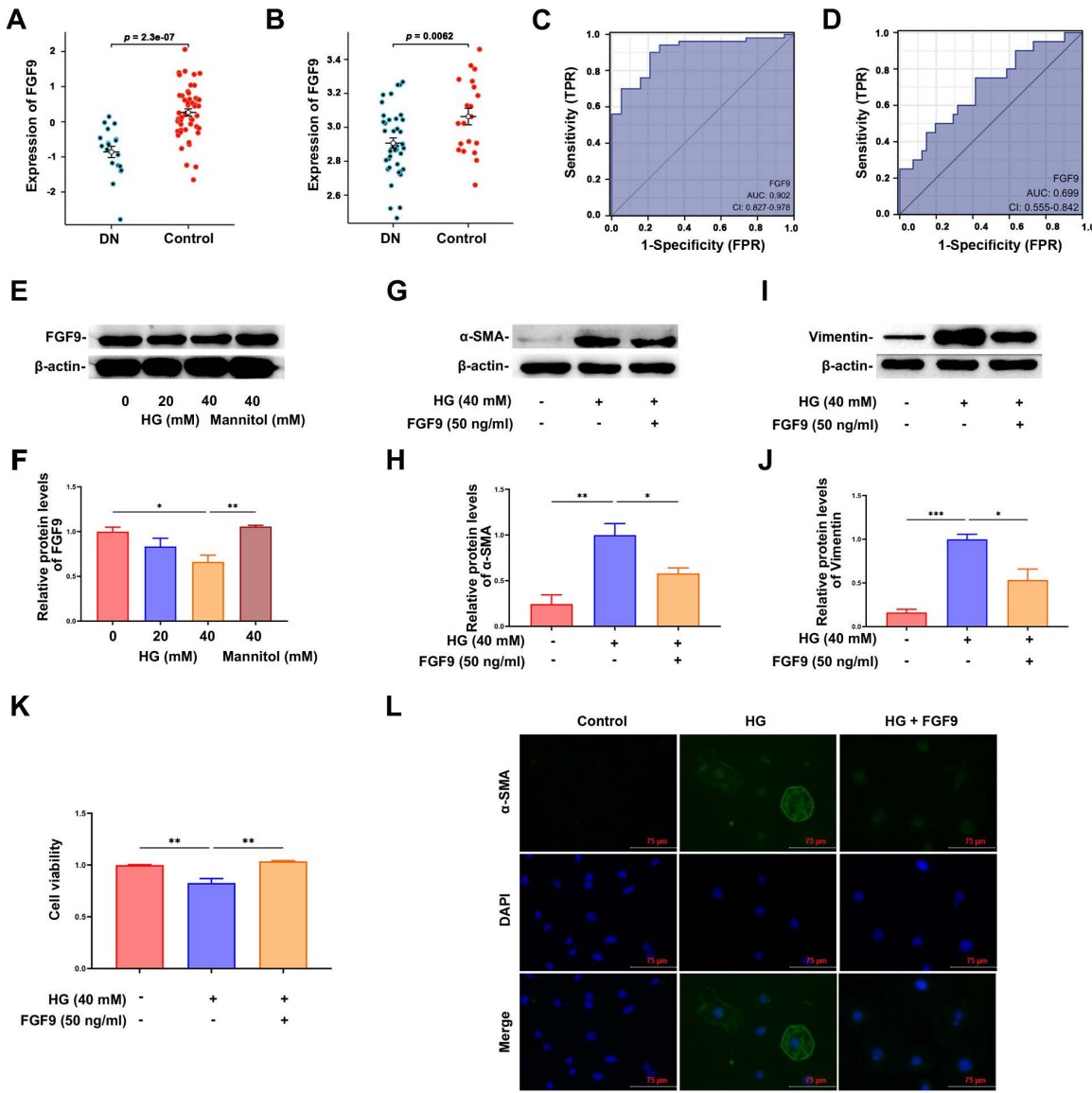

**Fig 8. Validation of FGF9 expression in external database and NRK-52E cells.** (A) The expression of FGF9 in the GSE30122 database was significantly decreased. (B) The expression of FGF9 in the GSE96804 database was significantly decreased. (C) AUC value of FGF9 in the GSE30122 database. (D) The AUC value of FGF9 in the GSE96804 database. (E) Representative western blots showed that FGF9 expression of NRK-52E under HG stimulation. (F) Summarized data of western blots showed the FGF9 expression was downregulated under HG stimulation. (G) Representative western blots showed that α-SMA protein expression after FGF9 treatment. (H) Summarized data of western blots showed that α-SMA protein expression was decreased after FGF9 treatment. (I) Representative western blots showed that vimentin protein expression after FGF9 treatment. (J) Summarized data of western blots showed that vimentin protein expression was decreased after FGF9 treatment. (K) Cell viability was measured by CCK8. (L) Representative images of immunofluorescence staining for a-SMA (green) and DAPI (blue) (Scale bars: 75 μm). * $p < 0.05$, ** $p < 0.01$, *** $p < 0.001$.

found that circRNA_15698 acted as a sponge for miR-185, inhibiting its expression and thereby increasing the expression of transforming growth factor-β1 protein, leading to the accumulation of glomerular extracellular matrix [47]. miRNAs in serum and urine exosomes have been shown to be associated with renal tubule interstitial injury. hsa-miR-34a-5p has been found to regulate oxidative stress damage and promote diabetic kidney damage [48]. hsa-miR-147 has been shown

to be upregulated in rats with doxorubicin-induced renal fibrosis [49]. The circulation hsa-27a-3p has been found to be involved in fibrosis in DN [50]. In the present study, four common miRNAs were obtained by intersecting the miRNAs specifically bound to differentially expressed circRNAs with the upstream miRNAs of FGF9. These were hsa-miR-34a-5p, hsa-miR-766-3p, hsa-miR-147a and hsa-miR-27a-3p. Respectively, the sponge circRNAs in serum exosomes were hsa-circ-0006382 and hsa-circ-0019539, based on which, the circRNA-miRNA-FGF9 ceRNA network was constructed.

FGF is one of the most diverse groups of growth factors in vertebrates [51]. The FGF9 subfamily is a member of the FGF families, including FGF9, 16 and 20, which have similar biochemical functions due to their high sequence homology [52]. The FGF9 signaling pathway is involved in the occurrence and development of multiorgan diseases [53,54]. To date, FGF9 has been extensively studied in heart disease. FGF9 signaling is a key mediator of myocardial growth and development, regulating coronary artery development. Exogenous FGF9 protein, or its specific expression in the heart under specific conditions, may improve function following myocardial infarction [55]. A previous study demonstrated that FGF9 attenuated myocardial infarction in diabetic mice by increasing the levels of anti-inflammatory cytokines and M2 macrophage differentiation, thereby reducing adverse remodeling and improving cardiac function [56]. FGF9 is selectively expressed in adults, including the central nervous system and kidneys [57]. However, the role of FGF9 in kidney disease has been rarely reported. In the present study, it was verified that HG stimulation decreased the expression of FGF9 in renal tubular epithelial cells in *in vitro* experiments. The results also revealed that the expression of FGF9 was downregulated though ceRNA mechanisms by hsa-circ-0006382 and hsa-circ-0019539, which were transported to the kidneys by circulating exosomes, exacerbating the EMT of the renal tubular epithelium.

The ERK signaling pathway played an important role in DN, especially in the progression of renal tubulointerstitial fibrosis. Studies had shown that ERK signaling pathway was directly involved in renal tubule fibrosis by regulating multiple processes such as cell proliferation, apoptosis and matrix deposition [58]. In the presence of high glucose, the ERK pathway of tubular epithelial cells was activated, and the ERK pathway was involved in the epithelial-mesenchymal transformation (EMT) process of tubular cells, which played a key role in the progression of DN [59]. Other studies had shown that FGF9 promotes neurite growth in Huntington's disease models through the ERK signaling pathway [60]. By binding to FGFR1, FGF9 activated FAK, AKT and ERK/MAPK signaling pathways to promote tumor cell proliferation and epithelial-mesenchymal transformation (EMT), thereby enhancing tumor cell migration and invasion [61]. However, whether FGF9 inhibited renal tubulointerstitial fibrosis through ERK pathways had not been investigated, and future studies will explore the role of these pathways.

## Conclusions

In conclusion, in the present study, the circRNA-miRNA-FGF9 mRNA network was constructed by high-throughput sequencing of circulating exosome-circRNAs, miRNA binding site prediction and WGCNA co-expression analysis, and the role of the key gene, FGF9, in the EMT of renal tubular epithelial cells was validated *in vitro*. The findings presented herein may provide new evidence of the role of the ceRNA mechanisms of the circRNA-miRNA interaction in renal fibrosis in DN. However, the clinical sample in this study was limited. The limited sample size might indeed have had an impact on the generality of the results. In order to eliminate the uncertainty brought by it, we further verified it through in vitro cell experiment and animal experiment. More samples would be collected in future studies to verify the generalizability of our results.

## Supporting information

**S1 Raw Images.**
(PDF)

**S1 File.**
(XLSX)

**S2 File. Raw data.**
(TIF)

## Author contributions

**Data curation:** Donglin YANG, Rongjiang Yin, Xiaomin Zhang, Xiaohui Wang, Xiaobin Pei, Zijie Guo, Pengyue Qiao, Kehan Zhu.

**Funding acquisition:** Pengchao Du.

**Methodology:** Xiaohui Wang, Xiaobin Pei, Zijie Guo, Pengyue Qiao, Kehan Zhu.

**Software:** Donglin YANG, Rongjiang Yin, Xiaomin Zhang.

**Supervision:** Lin Wang, Pengchao Du.

**Writing – original draft:** Donglin YANG.

**Writing – review & editing:** Lin Wang, Pengchao Du.

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
