## [Decision Letter · Decision Letter 0]

12 Mar 2025

Dear Dr. Du,

Thank you for submitting your manuscript to PLOS ONE. After careful consideration, we feel that it has merit but does not fully meet PLOS ONE’s publication criteria as it currently stands. Therefore, we invite you to submit a revised version of the manuscript that addresses the points raised during the review process.

We look forward to receiving your revised manuscript.

Kind regards,

Kai Huang

Academic Editor

PLOS ONE

“The present study was supported by Shandong Natural Science Fund of Shandong Province (grant no. ZR2020MH080); Introduction of Talent Research Initiation Fund of Central Hospital Affiliated to Shandong First Medical University (grant no. YJRC2022017); The Projects of Medical and Health Technology Development Program in Shandong Province (grant no. 202003050666, 2019WS310); The Projects of Technological Innovation Development Program in Yantai City (grant no.2021YD060, 2022YD071); Traditional Chinese medicine science and technology project of Shandong Province (grant no. M-2023017); Clinical +X project of Binzhou Medical University (BY2021LCX24).”

Reviewers' comments:

Reviewer's Responses to Questions

**Comments to the Author**

1. Is the manuscript technically sound, and do the data support the conclusions?

Reviewer #1: Yes

Reviewer #2: Yes

2. Has the statistical analysis been performed appropriately and rigorously?

Reviewer #1: No

Reviewer #2: Yes

3. Have the authors made all data underlying the findings in their manuscript fully available?

Reviewer #1: Yes

Reviewer #2: No

4. Is the manuscript presented in an intelligible fashion and written in standard English?

Reviewer #1: No

Reviewer #2: Yes

Reviewer #1: The manuscript by Yang et al. investigated the mechanistic role of exosome-circRNAs in the exacerbation of renal fibrosis in DN by downregulating the expression of FGF9 through ceRNA mechanism. The authors performed a comprehensive analysis by integrating both dry-lab and wet-lab data to support their claims. However, I found this manuscript had a major defect in their results presentation, which significantly make the results less convincing even though the figures have shown some promising findings. I explain my concerns in more details below.

Major comments:

1. The 'Abstract' section needs to be more concise as the current 'Abstract' described the background and methods in too many details, which will unavoidably make the topic of this study less prominent. I suggest the author shorten the 'Abstract' appropriately.

2. The English language of this manuscript needs significant improvement before it is suitable for publication.

3. Line 144, the author should use adjusted p-value for multiple testing correction. The same issue is applicable to the p-value < 0.05 in Line 168. Please indicate which corrected method when performing multiple testing correction.

4. In the ‘Results’ section, the results descriptions lack statistical data to supporting the findings. Especially for the first three paragraphs of the ‘Results’ section, I suggest the authors rewrite them. This is because the author actually only described 'what they have done in the study', but did not indicate 'what the results were'.

Minor comments:

5. Line 33, (DN) should be added to ‘diabetic nephropathy' in this sentence since DN abbreviation was used hereafter in the manuscript. On the contrary, Line 101, DN should be used rather than the full name for consistency purposes.

6. Lines 109-119, the author should provide the reference for the methods of exosome isolation. If no reference, the author should provide evidence that the exosome isolation process is successful in obtaining qualified exosomes for sequencing.

7. Line 282, the statistical data in the parentheses seems to be wrong here based on Figure5D. The authors should rewrite this sentence to make it correct.

8. Lines 307-308, I suggest the authors move this description to 'Methods' section and make some adjustment appropriately. It would be confusing to organize the manuscript elements in wrong sections.

9. The second paragraph of the ‘Discussion’ section should be moved to 'Introduction' section.

Reviewer #2: The manuscript investigates how circulating exosome-circRNAs mediate FGF9 downregulation via the ceRNA mechanism, exacerbating renal fibrosis in diabetic nephropathy (DN). The study uses high-throughput sequencing, bioinformatics analysis, and in vitro experiments to construct a circRNA-miRNA-FGF9 regulatory network, suggesting its role in fibrosis progression.

However, this paper is not good for publish because of the lack of sufficient validation and experimental depth.

1. The study relies on only six human samples (3 DN, 3 controls), which is too small for meaningful conclusions.

2. Functional validation is weak, focusing only on α-SMA expression and cell viability.

3. miRNA binding predictions lack experimental validation (e.g., overexpression/inhibition studies).

**Do you want your identity to be public for this peer review?** For information about this choice, including consent withdrawal, please see our Privacy Policy

Reviewer #1: **Yes: ** Jinpao Hou

Reviewer #2: No

---

## [Author Response · Author response to Decision Letter 1]

2 Apr 2025

Response to reviewers

We thank the editors and reviewers for the comments on this manuscript (PONE-D-25-08893). The concerns have been addressed as follows:

To Editor

1. Please ensure that your manuscript meets PLOS ONE's style requirements, including those for file naming. The PLOS ONE style templates can be found at https://journals.plos.org/plosone/s/file?id=wjVg/PLOSOne_formatting_sample_main_body.pdf and https://journals.plos.org/plosone/s/file?id=ba62/PLOSOne_formatting_sample_title_authors_affiliations.pdf.

Answer: Thank you very much for reminding. We have thoroughly reviewed our manuscript and ensured that it adheres to PLOS ONE's style requirements as outlined in the provided templates. All file names have been updated to comply with the specified guidelines.

2. Please state what role the funders took in the study. If the funders had no role, please state: "The funders had no role in study design, data collection and analysis, decision to publish, or preparation of the manuscript."If this statement is not correct you must amend it as needed.Please include this amended Role of Funder statement in your cover letter; we will change the online submission form on your behalf.

Answer: Thank you for your guidance regarding the role of the funders in our study. We have carefully reviewed the involvement of the funding sources and have determined that the funders had no role in study design, data collection and analysis, decision to publish, or preparation of the manuscript. We have included the following statement in our cover letter:

"The funders had no role in study design, data collection and analysis, decision to publish, or preparation of the manuscript. "

Answer: Thank you for your important reminder regarding the data availability statement. We appreciate your guidance on this matter. The data supporting our study is already publicly available. The data can be accessed through the following link: https://www.ncbi.nlm.nih.gov/sra/PRJNA1184123, or https://doi.org/10.6084/m9.figshare.28681061.

"Availability of data and materials: The sequencing data generated in the present study can be found in the National Center for Biotechnology Information under the login number of the data set: PRJNA1184123 or at the following URL: https://www.ncbi.nlm.nih.gov/sra/PRJNA1184123. All relevant data are within the paper and its Supporting Information files. The Supporting Information files have been uploaded to figshare. The direct link to freely access the Supporting Information files is https://doi.org/10.6084/m9.figshare.28681061. "

Answer: Thank you for your suggestion. We included the ethic statement only in Methods part..

" All human specimens were obtained from the Affiliated Hospital of Binzhou Medical University from April 22, 2021 to March 11, 2023. The patients were provided written informed consent to participate in this study. The six serum samples were from three healthy people and three DN patients. The criteria for inclusion of healthy people were disease-free. All experiments were all pathologically proven with an estimated glomerular filtration rate (eGFR) no less than 30 ml/min/ 1.73 m2 (based on the Chronic Kidney Disease Epidemiology Collaboration (CKD-EPI) equation). All experiments involving human subjects were approved by the Ethics Committee of Binzhou Medical University (protocol no. 2020-405). "

Answer: Thank you for your suggestion. We have provided all original uncropped and unadjusted images underlying the blot and gel results reported in our figures. These images are included in the Supporting Information files of our revised manuscript. In our cover letter, we have noted that all blot/gel image data are included in the Supporting Information. We have also provided specific details regarding the availability of each raw blot/gel image, confirming that there are no images that are not available.

Answer: Thank you for your suggestion. We have included captions for all Supporting Information files at the end of the manuscript and update any in-text citations to ensure they match the new format as per the Supporting Information guidelines provided by PLOS ONE.

Reviewer 1: The manuscript by Yang et al. investigated the mechanistic role of exosome-circRNAs in the exacerbation of renal fibrosis in DN by downregulating the expression of FGF9 through ceRNA mechanism. The authors performed a comprehensive analysis by integrating both dry-lab and wet-lab data to support their claims. However, I found this manuscript had a major defect in their results presentation, which significantly make the results less convincing even though the figures have shown some promising findings. I explain my concerns in more details below.

1. The 'Abstract' section needs to be more concise as the current 'Abstract' described the background and methods in too many details, which will unavoidably make the topic of this study less prominent. I suggest the author shorten the 'Abstract' appropriately.

Answer: Thank you for your suggestion. We have revised the "Abstrac" according to your suggestion.

" Diabetic nephropathy (DN) is one of the most serious microvascular complications of diabetes mellitus. It is characterized by progressive tubulointerstitial fibrosis. The aim of this study was to investigate the role of exosomal circular RNA (circRNAs) in regulating fibroblast growth factor 9 (FGF9) expression in DN through a competitive endogenous RNA (ceRNA) mechanism, and to reveal its potential therapeutic targets. Exosomes were isolated from serum of 3 healthy people and 3 patients with DN by ultra-fast centrifugation method, and the circRNA-miRNA-FGF9 regulatory network was constructed by combining high-throughput circRNA sequencing, bioinformatics analysis and weighted co-expression network (WGCNA). The results showed that the expression of circRNAs in serum exosomes of DN patients was significantly down-regulated, and hsa_circ_0006382 and hsa_circ_0019539 targeted the expression of FGF9 by binding to miR-34a-5p, miR-766-3p, miR-147a and miR-27a-3p. Further verification showed that the expression of FGF9 was decreased in renal tissues of DN patients (AUC=0.902), and its recombinant protein could inhibit the expression of α-SMA and vimentin in high glucose-induced NRK-52E cells, indicating that activation of the circRNA/miRNA-FGF9 network promotes the EMT of renal tubular epithelial cells. This study revealed for the first time the mechanism of the CircRNA-miRNA-FGF9 regulatory network in DN fibrosis, providing a theoretical basis for the development of diagnostic markers and targeted therapy strategies based on exosomal circRNA."

2. The English language of this manuscript needs significant improvement before it is suitable for publication.

Answer: Thanks for your suggestion, we have carried out the language polishing and obtained the language editing certificate. As follows:

3. Line 144, the author should use adjusted p-value for multiple testing correction. The same issue is applicable to the p-value < 0.05 in Line 168. Please indicate which corrected method when performing multiple testing correction.

Answer: Thank you very much for pointing out our mistake. We sincerely apologize for the inaccuracy in our description of the corrected P-value used in the circRNA differential expression analysis Method. In the figure, we used Padj.

The multiple hypothesis testing correction was performed to obtain the corrected P-value (Padj). The method used to calculate Padj was the Benjamini-Hochberg (BH) Method [1].

The set threshold of differential expression of circRNAs was | log2(Fold Change) | > 1 and Padj value < 0.05.

In the identification of differential expression analysis, the set threshold was |log2(Fold Change)| > 1 and Padj value < 0.05. The Padj was calculated using the Benjamini-Hochberg (BH) Method [1].

[1] Benjamini, Y., & Hochberg, Y. (1995). Controlling the false discovery rate: apractical and powerful approach to multiple testing. Journal of the Royalstatistical society: series B (Methodological), 57(1), 289-300.

We have revised the manuscript. As follows:

"According to the set threshold: | log2(Fold Change, FC) | > 1 and adjusted P- (Padj -) value < 0.05, circRNA differences between samples were screened. The corresponding Fold Change and Padj - value of up-regulated and down-regulated genes were obtained, and the volcano map was used for visualization. "

"The "limma" software package in R was used to identify differential expression genes (DEGs) between normal and DN samples, Padj < 0.05 and |log2FC| >1 gene was DEGs."

4. In the ‘Results’ section, the results descriptions lack statistical data to supporting the findings. Especially for the first three paragraphs of the ‘Results’ section, I suggest the authors rewrite them. This is because the author actually only described 'what they have done in the study', but did not indicate 'what the results were'.

Answer: Thank you for your suggestion. We have revised the first three paragraphs to include relevant statistical data according to your suggestion. And thank you for your careful suggestion again.

"Serum was obtained from 3 healthy individuals and 3 patients with DN, and exosomes were isolated and purified using the ultrafast centrifugation method. Transmission electron microscopy (TEM) was used to observe spherical particles with a diameter of 30-150 nm (Fig. 2A). The size of the particles analyzed by nanoparticle tracking analysis (NTA) technology was 81.35nm and the concentration was 1.93x109 particles/ml (Fig. 2B). The exosomes were extracted successfully. The purified exosomes were sequenced by circRNAs. The length distribution of circRNAs in each sample was statistically analyzed and plotted, the results showed that the length of most circRNAs was distributed between 149-15149nt (Fig. 2C). The type distribution map of the source regions of circRNAs showed circRNAs were derived from the splicing of exons, introns and intergenic regions (Fig. 2D). The chromosomal distribution map showed that the circRNA host genes were located on chromosome 1, 10, and 18. (Fig. 2E). The results of sample correlation heat map showed that the correlation between DN1 and DN2 was 0.997, indicating that the two samples were highly similar (Fig. 2F). The results of PCA diagram showed that the control group (red) and DN (blue) clustered on both sides of PC1 (46.97%), indicating that circRNA expression was significantly different between the two groups (Fig. 2G). The differential expression of circRNAs was analyzed, and the volcano map revealed that the expression of differentially expressed circRNAs was significantly decreased in DN (Fig. 2H). The clustering heatmap analysis revealed that circRNAs were generally green (low expression) in DN and red (high expression) in control group, indicating that low expression of circRNA might be related to DN (Fig. 2I). The top eight circRNAs were listed in descending order according to the | log2FC | value (Fig. 2J).

In order to further examine the biological function of differentially expressed circRNAs, GO, KEGG and Reactome were used to enrich the source genes of differentially expressed circleRNAs. Cellular component (CC) enrichment analysis revealed that these source genes were related to the N-terminal protein acetyltransferase complex, the site of DNA damage, the site of double-strand break, etc. Molecular function (MF) enrichment analysis revealed that these source genes were related to ATPase activator activity, RNA stem-loop binding, alpha-mannosidase activity, etc. (Fig. 3A). Similarly, KEGG pathway analysis revealed enrichment in protein processing in the endoplasmic reticulum (Padj < 0.01), porphyrin and chlorophyll metabolism (Padj < 0.05), N-Glycan biosynthesis (Padj < 0.05), Aminoacyl-tRNA biosynthesis (Padj < 0.05). (Fig. 3B). The results of Reactome enrichment analysis revealed that the circRNAs were related to tRNA modification in the nucleus and cytoplasm, and tRNA aminoacylation (Padj < 0.05), etc. (Fig. 3C).

The GSE30528 and GSE30529 dataset samples were combined after batch effect removal, including 19 DN tubule samples and 25 control samples. The data box plot after processing revealed that the median of each sample was almost on the same line, which meant the data were standardized successfully (Fig. 4A). PCA results showed PC1 (21%) and PC2 (16.9%), which indicated that samples were clustered according to DN and control group (Fig. 4B). The volcano map showed 202 upregulated genes and 64 downregulated genes in DN according to the analysis of DEGs (Fig. 4C). The heatmaps showed the top 10 upregulated (RARRES1, COL1A2, IGJ, IGLV1-44, LTF, C3, LYZ, ALOX5, CXCL6, MMP7) and downregulated genes (TYRP1, EGF, APOLD1, FOSB, NR4A3, ALB, ETNPPL, APOH, G6PC, CYP27B1) in DN (Fig. 4D)."

5. Line 33, (DN) should be added to ‘diabetic nephropathy' in this sentence since DN abbreviation was used hereafter in the manuscript. On the contrary, Line 101, DN should be used rather than the full name for consistency purposes.

Answer: Thank you for your reminding. According to your suggestion, we have added '(DN)' after 'diabetic nephropathy' in Line 33 and replaced the full name with 'DN' in Line 101 to ensure clarity and consistency throughout the manuscript.

6. Lines 109-119, the author should provide the reference for the methods of exosome isolation. If no reference, the author should provide evidence that the exosome isolation process is successful in obtaining qualified exosomes for sequencing.

Answer: Thank you for your suggestion. We have added references to the methods of exosome isolation in lines 109-119 of the manuscript.

Théry C, Amigorena S, Raposo G, Clayton A. Isolation and characterization of exosomes f

---

## [Decision Letter · Decision Letter 1]

28 May 2025

Circulating exosome-circRNAs mediated downregulation of FGF9 through ceRNA mechanism aggravates renal fibrosis in diabetic nephropathy.

PONE-D-25-08893R1

Dear Dr. Du,

We’re pleased to inform you that your manuscript has been judged scientifically suitable for publication and will be formally accepted for publication once it meets all outstanding technical requirements.

Kind regards,

Kai Huang

Academic Editor

PLOS ONE

Additional Editor Comments (optional):

Reviewers' comments:

Reviewer's Responses to Questions

**Comments to the Author**

Reviewer #1: All comments have been addressed

Reviewer #3: All comments have been addressed

2. Is the manuscript technically sound, and do the data support the conclusions?

Reviewer #1: Yes

Reviewer #3: Yes

3. Has the statistical analysis been performed appropriately and rigorously?

Reviewer #1: Yes

Reviewer #3: Yes

4. Have the authors made all data underlying the findings in their manuscript fully available?

Reviewer #1: Yes

Reviewer #3: Yes

5. Is the manuscript presented in an intelligible fashion and written in standard English?

Reviewer #1: Yes

Reviewer #3: Yes

Reviewer #1: I have reviewed all of the responses from the authors and find that the authors have carefully addressed all of my comments. I have no further comments.

Reviewer #3: The authors have thoroughly addressed all of my previous concerns, and the revised manuscript now meets the expectations for publication.

**Do you want your identity to be public for this peer review?** For information about this choice, including consent withdrawal, please see our Privacy Policy

Reviewer #1: **Yes: ** Jinpao Hou

Reviewer #3: **Yes: ** Minghan Chi

---

## [Editor Report · Acceptance letter]

PONE-D-25-08893R1

PLOS ONE

Dear Dr. Du,

I'm pleased to inform you that your manuscript has been deemed suitable for publication in PLOS ONE. Congratulations! Your manuscript is now being handed over to our production team.

Kind regards,

on behalf of

Dr. Kai Huang

Academic Editor

PLOS ONE